# Secure Quantized Training for Deep Learning

## Abstract

We have implemented training of neural networks in secure multi-party computation (MPC) using quantization commonly used in said setting. To the best of our knowledge, we are the first to present training of MNIST purely implemented in MPC that comes within one percent of accuracy of training using plaintext computation. We found that training with MPC is possible, but it takes more epochs and achieves a lower accuracy than the usual CPU/GPU computation. More concretely, we have trained a network with two convolution and two dense layers to 98.5% accuracy in 150 epochs. This took a day in our MPC implementation.

## 1  Introduction

Secure multi-party computation (MPC) is a cryptographic technique that allows a set of parties to compute a public output on private inputs without revealing the inputs or any intermediate results. This makes it a potential solution to federated learning where the sample data stays private and only the model or even only inference results are revealed.

Imagine a set of healthcare providers holding sensitive patient data. MPC allows them to collaboratively train a model. This model could then either be released or even kept private for inference using MPC again. See Figure for an illustration. A more conceptual example is the well-known millionaires' problem where two people want to find out who is richer without revealing their wealth. There is clearly a difference between the one bit of information desired and the full figures.

There has been a sustained interest in applying secure computation to machine learning and neural networks going back to at least Barni et al. [2006]. More recent advantages in practical MPC have led to an increased effort in implementing both inference and training.

A number of works such as Mohassel and Zhang [2017], Mohassel and Rindal [2018], Wagh et al. [2019], Wagh et al. [2021] implement neural network training with MPC at least in parts. However, they either give accuracy figures below 95% or figures that have been obtained using plaintext training. For the latter case, the works do not clarify how close the computation for plaintext training matches the lower precision and other differences in the MPC setting. Agrawal et al. [2019] claim a higher accuracy in a comparable setting for a convolutional neural network with more channels than we use. However, they have only implemented dense layers, and we achieve comparable accuracy to them with only dense layers. All works use quantization in the sense that a fractional number $x$ is represented as $\lfloor x \cdot 2^{-f} \rceil$. This makes addition considerably faster in the secure computation setting because it reduces to integer addition. Furthermore, some of the works suggest to replace the softmax function that uses exponentiation with a ReLU-based replacement. Keller and Sun [2020] have found that this softmax replacement deteriorates accuracy in dense neural networks to the extent that it does not justify the performance gains.

The concurrent work of Tan et al. [2021] gives some figures on the learning curve when run using secure computation. However, they stop at five epochs for MNIST training where they achieve 94% accuracy whereas we present the figures up to 150 epochs and 98.5% accuracy. Furthermore, their

Submitted to 35th Conference on Neural Information Processing Systems (NeurIPS 2021). Do not distribute.

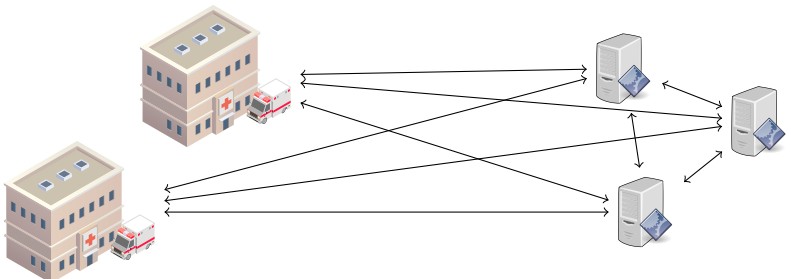

Figure 1: Outsourced computation: Data holders (on the left) secret-share their data to a number of computing parties (on the right), who then return the desired the result (e.g., a model or inference results on further queries). All communication except outputs are secret-shared and thus secure if no two computing parties collude.

choice of fixed-point precision 20 is considerably below 32, which we found to be optimal. We also found that our implementation is 40% faster than theirs. Note that we use the CPU of one AWS c5.9xlarge instance per party whereas Tan et al. use one NVIDIA Tesla V100 GPU per party. We believe this somewhat counter-intuitive result comes from MPC heavily relying on communication, which is an aspect where GPUs do not have an advantage over CPUs.

In this paper, we present an extensible framework for implementing deep learning training based on MP-SPDZ by Keller [2020], a framework for multi-party computation.[1] Similar to TensorFlow and PyTorch, our approach allows representing deep learning models as succession of layers. We then use this implementation to obtain accuracy figures for MNIST training by utilizing the MP-SPDZ emulator, which allows to run the plaintext equivalent of secure computation, that is, the same algorithms with the same precision. Finally, we run one of the most promising instantiation in real secure computation in order to benchmark it confirm the result from the plaintext emulator.

There are a number of projects that integrate secure computation directly into popular machine learning frameworks such as CrypTen by Gunning et al. [2019], PySyft by Ryffel et al. [2018], and TF Encrypted by Dahl et al. [2018]. Our approach differs from all of them by running the protocol as native CPU code (implemented using C++). This allows for much faster execution. For example, CrypTen provides an MNIST training example (`mpc_autograd_cnn`) that takes over one minute to run one epoch with 100 samples on one machine. In comparison, our implementation takes 11 minutes to run one epoch with the full dataset of 60,000 samples.

Another line of work (e.g., Quoc et al. [2021]) uses trusted execution environments that provide computation outside the reach of the operating system. This is a different security model to multi-party computation that works with distributing the information among several entities.

The paper is structured as follows: After introducing the basics of the protocol we use in Section 2, we will explain the mathematical building blocks in Section 3 and their use in the context of deep learning in Section 4. Finally, we will present our implementation in Section 5 and our experimental results for MNIST in Section 6.

## 2 An Efficient Secure Multi-Party Computation Protocol

There is a wide range of MPC protocols with a variety of security properties (see Keller [2020] for an overview). In this paper we focus on the setting of three-party computation with one semi-honest corruption. This means that out of the three parties two are expected to behave honestly, i.e., they follow the protocol and keep their view of the protocol secret, and one party is expected to follow the protocol but might try extract information from their view. The reason for choosing this setting is that it allows an efficient MPC protocol while still allowing secure outsourced computation. The concrete protocol we use goes back to Benaloh and Leichter [1990] with further aspects by Araki et al. [2016], Mohassel and Rindal [2018], and Eerikson et al. [2020]. We summarize the core protocol below. The mathematical building blocks in the next section mostly use the aspects below.

---

[1]We are committed to publishing our code as open source.

**Secret sharing** All intermediate values in our protocol are stored using replicated secret sharing. A secret value $x$ is a represented as a random sum $x = x_0 + x_1 + x_2$, and party $P_i$ holds $(x_{i-1}, x_{i+1})$ where the indices are computed modulo three. Clearly, each party is missing one value to compute the sum. On the other hand, each pair of parties hold all necessary to reconstruct the secret. For a uniformly random generation of shares, the computation domain has to be finite. Most commonly, this domain is defined by integer computation modulo a number. We use $2^k$ for $k$ being a multiple 64 and 2 as the moduli. The first case corresponds to an extension of 64-bit arithmetic found on most processors. We will refer to the two settings as arithmetic and binary secret sharing throughout the paper.

**Input sharing** The secret sharing scheme implies a protocol to share inputs where the inputting party samples the shares and distributes them accordingly. Eerikson et al. [2020] have proposed a more efficient protocol where the inputting party only needs to send one value instead of two pairs of values. If $P_i$ would like to input $x$, $x_i$ is set to zero, and $x_{i-1}$ is generated with a pseudo-random generator using a key previously shared between $P_i$ and $P_{i+1}$. $P_i$ can compute $x_{i+1} = x - x_{i-1}$ and send it to $P_{i-1}$. While the resulting secret sharing is not entirely random, the fact that $P_i$ already knows $x$ makes randomizing $x_i$ obsolete.

**Addition** The commutative nature of addition allows to add secret sharings without communication. More concretely, secret sharings $x = x_0 + x_1 + x_2$ and $y = y_0 + y_1 + y_2$ imply the secret sharing $x + y = (x_0 + y_0) + (x_1 + y_1) + (x_2 + y_2)$.

**Multiplication** The product of $x = x_0 + x_1 + x_2$ and $y = y_0 + y_1 + y_2$ is

$$x \cdot y = (x_0 + x_1 + x_2) \cdot (y_0 + y_1 + y_2)$$
$$= (x_0 y_0 + x_0 y_1 + x_1 y_0) + (x_1 y_1 + x_1 y_2 + x_1 y_1) + (x_2 y_2 + x_2 y_0 + x_0 y_2).$$

Each of the brackets only contains shares known by one of the parties. They can thus compute an additive secret sharing (one summand per party) of the product. However, every party only holding one share does not satisfy the replication requirement for further multiplications. It is not secure for every party to pass their value on to another party because the summands are not distributed randomly. This can be fixed by rerandomization: Let $xy = z_0 + z_1 + z_2$ where $z_i$ is know to $P_i$. Every party $P_i$ computes $z_i' = z_i + r_{i,i+1} - r_{i-1,i}$ where $r_{i,i+1}$ is generated with a pseudo-random generator using a key pre-shared between $P_i$ and $P_{i+1}$. The resulting sum $xy = z_0' + z_1' + z_2'$ is pseudo-random, and it is thus secure for $P_i$ to send $z_i'$ to $P_{i+1}$ in order to create a replicated secret sharing $((xy)_{i-1}, (xy)_{i+1}) = (z_i', z_{i-1}')$.

# 3 Secure Computation Building Blocks

In this section, we will discuss how to implement computation with MPC with a focus on how it differs from computation on CPUs or GPUs. Most of the techniques below are already known individually. To the best of our knowledge however, we are the first to put them together in an efficient and extensible framework for secure computation of deep learning training.

**Domain conversion** Recall we that we use computation modulo $2^k$ for $k$ being a multiple of 64 as well as 1. Given that the main operations are just addition and multiplication in the respective domain, it is desirable to compute integer arithmetic in the large domain but operations with a straight-forward binary circuit modulo two. There has been a long-running interest in this going back to least Kolesnikov et al. [2013]. We mainly rely on the approach proposed by Mohassel and Rindal [2018] and Araki et al. [2018]. Recall that $x \in 2^k$ is shared as $x = x_0 + x_1 + x_2$. Now let $\{x_0^{(i)}\}_{j=0}^{k-1}$ the bit decomposition of $x_0$, that is, $x_0^{(i)} \in \{0, 1\}$ and $x_0 = \sum_{i=0}^{k-1} x_0^{(i)} 2^i$. It is self-evident that $x_0^{(i)} = x_0^{(i)} + 0 + 0$ is a valid secret sharing modulo two (albeit not a random one). Furthermore, every party holding $x_0$ can generate $x_0^{(i)}$. It is therefore possible for the parties to generate a secret sharing modulo two of a *single share* modulo $2^k$. Repeating this for all shares and the computing the addition as a binary circuit allows the parties to generate a secret sharing modulo two from a secret sharing modulo $2^k$. Conversion in the other direction can be achieved using a similar technique or using daBits as described by Rotaru and Wood [2019]. In the following we will use the term mixed-circuit computation for any technique that works over both computation domains.

**Quantization** While Aliasgari et al. [2013] showed that it is possible to implement floating-point computation, the cost is far higher than integer computation. It is therefore common to represent fractional numbers using quantization (also called fixed-point representation) as suggested by Catrina and Saxena [2010]. A real number $x$ is represented as $\tilde{x} = \lfloor x \cdot 2^f \rceil$ where $f$ is an integer specifying the precision. The linearity of the representation allows to compute addition by simply adding the representing integers. Multiplication however requires to adjust the result because it will have twice the precision: $(x \cdot 2^f) \cdot (y \cdot 2^f) = xy \cdot 2^{2f}$. There are two ways to rectify this:

- An obvious correction would be to shift the result by $f$ bits after adding $2^{f-1}$ to the integer representation. This ensures rounding to the nearest number possible in the representation, with the tie being broken by rounding up. Dalskov et al. [2021] presented an efficient implementation of the truncation using mixed-circuit computation.

- However, Catrina and Saxena have found that in the context of secure computation it is more efficient to use probabilistic truncation. This method rounds up or down probabilistically depending on the input. For example, probabilistically rounding 0.75 to an integer would see it rounded of up with probability 0.75 and down with probability 0.25. The probabilistic truncation is an effect of the fact that the operation involves the truncation of a randomized value, that is the computation of $\lfloor (x + r)/2^m \rfloor$ for a random $m$-bit value $r$. It is easy to see that

$$\lfloor (x + r)/2^m \rfloor = \begin{cases} \lfloor x/2^m \rfloor & (x \bmod 2^m + r) < 2^m \\ \lfloor x/2^m \rfloor + 1 & (x \bmod 2^m + r) \geq 2^m. \end{cases}$$

  Therefore, the larger $(x \bmod 2^m)$ is, the more likely the latter condition is true. Dalskov et al. [2020] present an efficient protocol in our security model.

Our quantization scheme is related to quantized neural networks (see e.g. Hubara et al. [2016]). However, our consideration is not to compress the model, but to improve the computational speed and save communication cost.

**Dot products** Dot products are an essential building block of linear computation such as matrix multiplication. In the light of quantization, it is possible to reduce the usage of truncation by deferring after the summation. In other words, the dot product in the integer representations is computed before truncating. This not only reduces the truncation error, it is also more efficient because the truncation is the most expensive part in quantized secure multiplication. Similarly, our protocol allows to defer the communication needed for multiplication. Let $\vec{x}$ and $\vec{y}$ be two vectors where the elements are secret shared, that is, $\{x^{(i)}\} = x_0^{(i)} + x_1^{(i)} + x_2^{(i)}$ and similarly for $y^{(i)}$. The inner product then is

$$\sum_i x^{(i)} \cdot y^{(i)} = \sum_i (x_0^{(i)} + x_1^{(i)} + x_2^{(i)}) \cdot (y_0^{(i)} + y_1^{(i)} + y_2^{(i)})$$

$$= \sum_i (x_0^{(i)} y_0^{(i)} + x_0^{(i)} y_1^{(i)} + x_1^{(i)} y_0^{(i)}) + \sum_i (x_1^{(i)} y_1^{(i)} + x_1^{(i)} y_2^{(i)} + x_1^{(i)} y_1^{(i)})$$

$$+ \sum_i (x_2^{(i)} y_2^{(i)} + x_2^{(i)} y_0^{(i)} + x_0^{(i)} y_2^{(i)}).$$

The three sums in the last term can be compute locally by one party each before applying the same protocol as for a single multiplication.

**Comparisons** Arithmetic secret sharing does not allow to access the individual bits directly. It is therefore not straightforward to compute comparisons such as "less than". There is a long line of literature on how to achieve this going back to at least Damgård et al. [2006]. More recently, most attention has been given to combine the power of arithmetic and binary secret sharing in order to combine the best of worlds. One possibility to do so is to plainly convert to the binary domain and compute the comparison circuit there. In our concrete implementation we use the more efficient approach by Mohassel and Rindal [2018]. It starts by taking the difference of the two inputs. Computing the comparison then reduces to comparing to zero, which in turn is equivalent to extracting the most significant bit as it indicates the sign. The latter is achieved by converting the shares locally to bit-wise sharing of the arithmetic shares, which sum up to the secret value. It remains to compute the sum of the binary shares in order to come up with the most significant bit.

**Oblivious Selection**  Plain secure computation does not allow branching because the parties would need to be aware which branch is followed. Conditional assignment can be implemented as follows however. If $b \in \{0, 1\}$ denotes the condition, $x + b \cdot (y - x)$ is either $x$ or $y$ depending on $b$. If the condition is available in binary secret sharing but $x$ and $y$ in arithmetic secret sharing, $b$ has to be converted to the latter. This can be done using a daBit as introduced by Rotaru and Wood [2019], which is a secret random bit shared both in arithmetic and binary. It allows to mask a bit in one world by XORing it. The result is then revealed and the masking is undone in the other world.

**Division**  Catrina and Saxena [2010] have shown how to implement quantized division using the algorithm by Goldschmidt [1964]. It mainly uses arithmetic and the probabilistic truncation already explained. In addition, the initial approximation requires a full bit decomposition as described above. The error of the output depends on the error in the multiplications used for Goldschmidt's iteration, which compounds in particular when using probabilistic truncation. Due to the nature of secure computation, the result of division by zero is undefined. One could obtain a secret failure bit by testing the divisor to zero. However, we found that not to be necessary in our algorithm. This is because we only use division by secret value only for the softmax function where the it is guaranteed to strictly positive.

**Logarithm**  Computation logarithm with any public base can be reduced to logarithm to base two using $\log_x y = \log_2 y \cdot \log_x 2$. Aly and Smart [2019] have proposed computing $y = a \cdot 2^b$ where $a \in [0.5, 1)$ and $b \in \mathbb{Z}$. This then allows to compute $\log_2 y = \log_2 a + b$. Given the restricted range of $a$, $\log_2 a$ can be approximated using a division of polynomials. Numerical stability and input range control are less of an issue here because we only use logarithm for the loss computation, which does not influence the training.

**Exponentiation**  By using $x^y = 2^{y \log_2 x}$, any exponentiation can be reduced to exponentiation with base two. Aly and Smart [2019] have shown how to compute $2^a = 2^{\lfloor a \rfloor} \cdot 2^{a - \lfloor a \rfloor}$ by computing the two exponents using bit decomposition and the second factor using a polynomial approximation. Regarding the first factor, if $b = \sum_{i \geq 0} b_i 2^i$ is an integer with $b_i \in \{0, 1\}$,

$$2^b = 2^{\sum_{i \geq 0} b_i 2^i} = \prod_{i \geq 0} 2^{b_i 2^i} = \prod_{i \geq 0} (1 + b_i \cdot (2^{2^i} - 1)).$$

As with division the numerical stability depends on the truncation used for multiplication.

**Inverse square root**  Aly and Smart [2019] have proposed to compute square root using Goldschmidt and Raphson-Newton iterations. We could combine this with the division above. However, Lu et al. [2020] have proposed a more direct computation that avoids running two successive iterations.

**Uniformly random fractional number**  Limiting ourselves to intervals of the form $[x, x + 2^e]$ for a potentially negative integer $e$, we can reduce the problem to generate a random $(f + e)$-bit number where $f$ is the fixed-point precision. Recall that we represent a fractional number $x$ as $\lfloor x \cdot 2^{-f} \rceil$. Generating a random $n$-bit number is straight-forward using random bits, which in our protocol can be generated as presented by Damgård et al. [2019]. In the context of our protocol however, Dalskov et al. [2021] have a presented a more efficient approach that involves mixed-circuit computation.

**Communication cost**  Table 1 show the total communication cost of some of the building blocks in our protocol for $f = 32$. This setting mandates the modulus $2^{128}$ because the division protocol requires bit length $4f$.

# 4  Machine Learning Building Blocks

In this section, we will use the building blocks in Section 3 to construct high-level computational modules for deep learning.

**Fully connected layers**  Both forward and back-propagation of fully connected layers can be seen as matrix multiplications and thus can be implemented using dot products. A particular challenge in secure computation is to compute a number of outputs in parallel in order to save communication

Table 1: Communication cost of select computation for $f = 32$ and integer modulus $2^{128}$.

|  | Bits |
|---|---|
| Integer multiplication | 384 |
| Probabilistic truncation | 1,536 |
| Nearest truncation | 4,462 |
| Comparison | 1,369 |
| Division (prob. truncation) | 29,866 |
| Division (nearest truncation) | 57,798 |
| Exponentiation (prob. truncation) | 77,684 |
| Exponentiation (nearest truncation) | 171,638 |
| Invert square root (prob. truncation) | 20,073 |
| Invert square root (nearest truncation) | 27,699 |

rounds. We solve this by having a dedicated infrastructure in our implementation that computes all dot products for a matrix multiplication in a single batch, thus combining all necessary communication.

**2D convolution layers**  Similar to fully connected layers, 2D convolution and its corresponding gradient can be implemented using only dot products, and we again compute several output values in parallel.

**Rectified Linear Unit (ReLU)**  ReLU Nair and Hinton [2010] is defined as follows:

$$\mathsf{ReLU}(x) := \begin{cases} x, & \text{if } x > 0 \\ 0. & \text{otherwise} \end{cases}$$

It can thus be implemented as a comparison followed by an oblivious selection. For back-propagation, it is advantageous to reuse the comparison results from forward propagation due to the relatively high cost in secure computation. Note that the comparison results are stored in secret-shared form and thus there is no reduction in security.

**Max pooling**  Similar to ReLU, max pooling can be reduced to comparison and oblivious selection. In secure computation, it saves communication rounds if the process uses a balanced tree rather than iterating over all input values of one maximum computation. For back-propagation it again pays off to the store intermediate results from forward propagation, again in secret-shared form.

**Softmax and cross entropy loss**  This combination requires computing the following gradient for back-propagation:

$$\bigtriangledown_i := \frac{\partial \ell}{\partial x_i} = \frac{\partial}{\partial x_i}\Big( -\sum_k y_k \cdot x_k + \log \sum_j e^{x_j} \Big) = -y_i + \frac{e^{x_i}}{\sum_j e^{x_j}}, \tag{1}$$

where $y_i$ denotes an element of the ground truth as a one-hot vector, and $x_i$ denotes the output of the last layer.

On the right hand side of eq. (1), the values in the denominator are potentially large due to the use of the exponential. This is prone to numerical overflow in our quantized representation because the latter puts relatively strict limits on the values. We therefore optimize the computation by computing the maximum of the input values:

$$m = \max_j(\{x_j\}).$$

Then we compute

$$\frac{e^{x_i - m}}{\sum_j e^{x_j - m}} = \frac{e^{x_i} e^{-m}}{(\sum_j e^{x_j}) e^{-m}} = \frac{e^{x_i}}{\sum_j e^{x_j}}.$$

All the exponents on the left-most term are at most zero, and thus the dividend is at most one and the divisor is at most the number of labels (which is 10 in MNIST). The same technique can be used to compute the sigmoid activation function, as $\mathrm{sigmoid}(x) = \frac{1}{1+\exp(-x)} = \frac{\exp(0)}{\exp(0)+\exp(x)}$ is a special case of softmax.

**Stochastic gradient descent**    The model parameter update in SGD only involves basic arithmetic: $\theta_j \leftarrow \theta_j - \frac{\gamma}{B} \sum_{i=1}^{B} \bigtriangledown_{ij}$ where $\theta_j$ is the parameter indexed by $j$, $B$ is the mini-batch size, $\gamma > 0$ is the learning rate, and $\bigtriangledown_{ij}$ is the gradient of the loss with respect to the $i$'th sample in the mini-batch and the parameter $\theta_j$. In order to tackle the limited precision with quantization, we defer dividing by the batch size to the model update. This means that we do not divide the gradient value by the batch size when computing them as described in the previous paragraph. Instead, we divide the model update by the batch size. Since we use a batch size that is a power of two (128), it is sufficient to use probabilistic truncation instead of full-blown division. This saves both time and decreases the error.

**Adam**    The main difference to SGD in terms of basic computational operations is the additional use of an inverse square root. We again defer the division by the batch size to just before the model update.

**Parameter initialization**    We use the Glorot initialization by Glorot and Bengio [2010]. Besides basic operations, it mainly involves generating a uniformly random fractional value in a given interval.

# 5   Implementation

We built our implementation on MP-SPDZ by Keller [2020]. MP-SPDZ not only implements a range of MPC protocols, it also comes with a high-level library containing the building blocks in Section 3. MP-SPDZ already featured capabilities to train dense neural networks as well as inference for convolutional neural networks. We have added backward propagation for a number of layer types, including 2D convolution. Furthermore, we have corrected a bug in the backward propagation for dense layers.

MP-SPDZ allows implementing the computation in Python code, which is then compiled a specific bytecode. This code can be execute by a virtual machine executing the actual secure computation. The process allows to optimize the computation in the context of MPC.

The framework also features an emulator that executes the exact computation that could be done securely in the clear. This allowed us to collect the accuracy figures in the next section at a lower cost.

It is licensed under a BSD-style license, which allows to extend the code.

# 6   MNIST

For a concrete measurement of accuracy and running times, we have implemented training for the well-known MNIST dataset by LeCun et al. [2010]. We work mainly with the models that have been used by Wagh et al. [2019] with secure computation, and we will reuse their numbering (A–D). The models contain up to four linear layers. Network C is a convolutional neural network going back to the seminal work by LeCun et al. [1998] whereas the others are simpler networks that have been proposed by works on secure computation such as Mohassel and Zhang [2017], Liu et al. [2017], and Riazi et al. [2018]. We present the networks as Keras code in the supplemental material.

Figure 2 shows the results for various quantization precisions and and the two rounding options. We have used SGD with learning rate 0.01, batch size 128, and the usual MNIST training/test split. $f = 64$ is the best option with probabilistic rounding, improving on both $f = 16$ and $f = 32$. Furthermore, nearest rounding performs worse that probabilistic for $f = 16$ and $f = 32$. Due to the high cost, we only ran $f = 32$ with probabilistic rounding several times. The range is indicate by the shaded area. We focus on $f = 32$ because it offers the faster convergence.

Figure 3 then shows the result with a variety of optimizers. While increasing the learning rate for SGD leads to a lower stability, Adam exposes a smoother learning learning curve albeit not a faster process.

Finally, Figures 4 shows our results for all networks used by Wagh et al. [2019]. As one would expect, the most sophisticated network performs best. Somewhat surprisingly, however, Network A (without convolutional layers) performs better than the simpler networks containing convolutional layers.

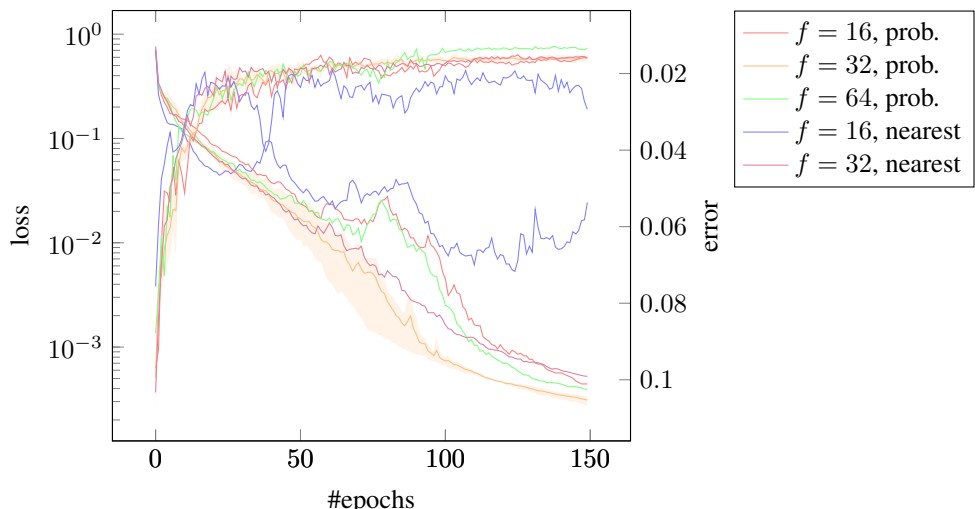

Figure 2: Loss and accuracy for network C and precision options when running SGD with rate 0.01.

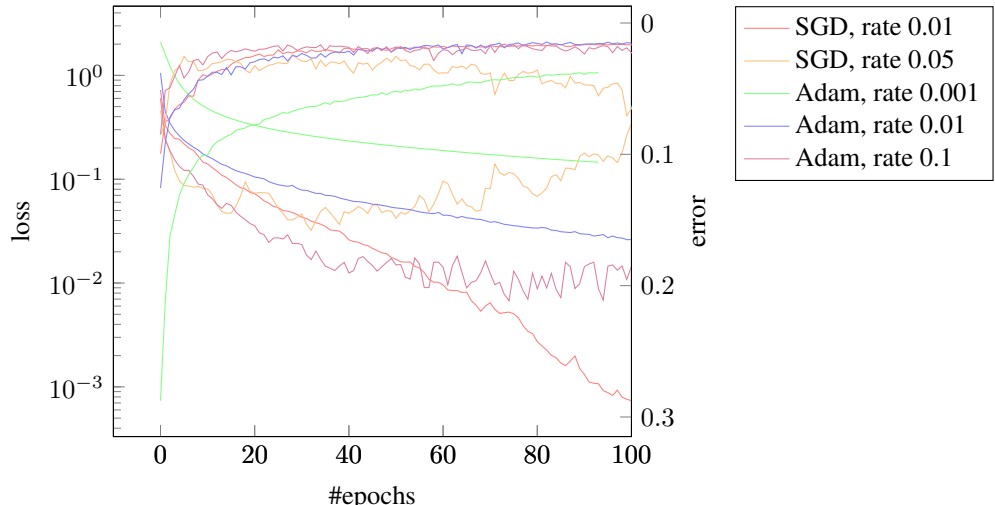

Figure 3: Loss and accuracy for network C with various optimizer options, $f = 32$, and probabilistic truncation.

**Resources** We ran the emulator on AWS `c5.9xlarge` instances. One epoch takes a few second to several minutes depending on the model. Overall, we estimate that we have used a few weeks worth of computing time including experiments not included here because of bugs in the code.

## 6.1 Secure computation

In order to verify our emulation results, we have run Network C with precision $f = 32$ and probabilistic rounding in our actual multi-party computation protocol. We could verify that it converges on 98.5% accuracy at 150 epochs, taking 20 hours. Table 2 compares our result to previous works in a LAN setting. Note that Wagh et al. [2019] and Wagh et al. [2021] give accuracy figures. From personal communication with the authors and the fact that the source repository for the latter work[2] says that their "code has not run end-to-end training", we derive our assessment that their figures do not reflect the secure computation.

---

[2]`https://github.com/snwagh/falcon-public`

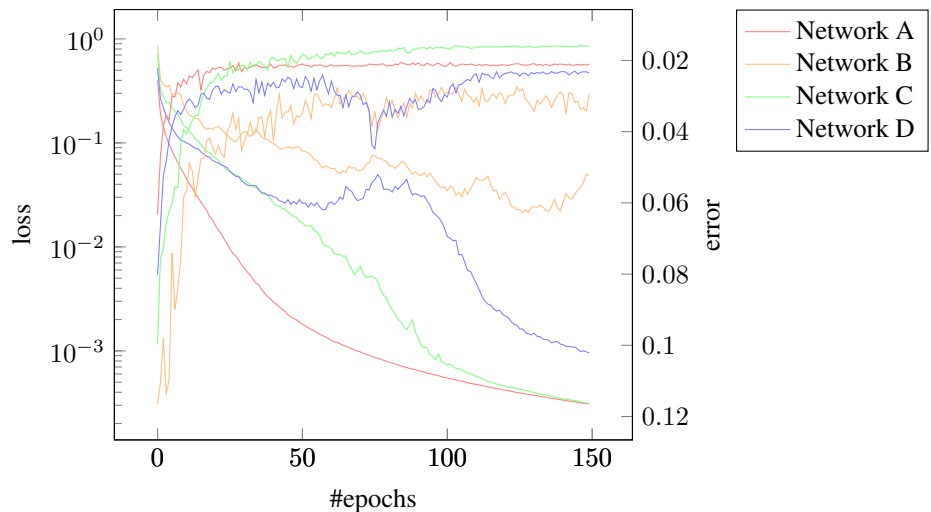

Figure 4: Loss and accuracy for various networks, $f = 32$, and probabilistic truncation.

Table 2: Comparison to previous work in the LAN setting. ($^*$) Mohassel and Zhang [2017] and Agrawal et al. [2019] use a different security model and are thus incomparable. We include them for completeness. Two numbers refer to online and offline time. Accuracy N/A means that the accuracy figures were not given or computed in a way that does not reflect the secure computation.

| Network | | Epoch time (s) | Acc. (# epochs) | Precision ($f$) |
|---|---|---|---|---|
| A | Mohassel and Zhang [2017] | 283/19333$^*$ | 93.4% (15) | 13 |
| | Mohassel and Rindal [2018] | 180 | 94.0% (15) | N/A |
| | Agrawal et al. [2019] | 31392$^*$ | 95.0% (10) | N/A |
| | Wagh et al. [2019] | 247 | N/A | 13 |
| | Wagh et al. [2021] | 41 | N/A | 13 |
| | Ours | 31 | 97.9% (15) | 16 |
| | Ours | 50 | 97.7% (15) | 32 |
| B | Wagh et al. [2019] | 4176 | N/A | 13 |
| | Wagh et al. [2021] | 101 | N/A | 13 |
| | Ours | 144 | 93.6% (15) | 16 |
| | Ours | 249 | 94.7% (15) | 32 |
| C | Wagh et al. [2019] | 7188 | N/A | 13 |
| | Wagh et al. [2021] | 891 | N/A | 13 |
| | Tan et al. [2021] | 1036 | 94.0% (5) | 20 |
| | Ours | 344 | 94.9% (5) | 16 |
| | Ours | 643 | 93.8% (5) | 32 |
| D | Mohassel and Rindal [2018] | 234 | N/A | N/A |
| | Ours | 41 | 96.8% (15) | 16 |
| | Ours | 68 | 96.8% (15) | 32 |

## 7  Conclusions

We have presented an implementation of deep learning training purely in multi-party computation with extensive results on the accuracy. We have found that the lower precision of MPC computation increases the error considerably. We only have considered one particular implementation of more complex computation such as division and exponentiation, which are crucial to the learning process as part of softmax. Future work might consider different approximations of these building blocks.

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
