# OpenReview forum: "Secure Quantized Training for Deep Learning"
_NeurIPS.cc/2021/Conference — NeurIPS 2021 Submitted_

### Official Review · Reviewer_Sgcg · 2021-07-13

**Rating:** 3
**Confidence:** 4

**Summary:**

Summary: This paper proposes an implementation and evaluation of an MPC protocol for training an MNIST model. The trained model achieves accuracy above 98% in 1 day.

**Ethics Review Area:**

["I don’t know"]

**Limitations And Societal Impact:**

-

**Main Review:**

This paper aims to push the boundaries of practical neural network training with MPC.
The paper does not present any new techniques, but rather implements a collection of standard approaches from prior work and then evaluates a small number of model architectures and parameter configurations on MNIST.

From the introduction, it is hard to extract what this paper claims as its main contribution.
After defining MPC, the introduction starts discussing some intricate details about prior work without giving much context. It then follows with some low-level implementation details (e.g., the type of CPU or GPU used in prior work).
After that, we seem to get to the real contribution: a framework for implemnting MPC models.
But here again, it isn't very clear what this framework presumably does (or allows to do) that prior work didn't.

A very large part of the rest of the paper (p.3-6) presents known results from the literature, at times in overly detailed manner. In particular, it wouldn't be unnatural to assume that any NeurIPS reader is intimately familiar with the entirety of Section 4.

The framework that is claimed as the main contribution is never described in detail. What exactly does this paper propose that couldn't be done with prior tools? Or is the main contribution the engineering effort of porting existing techniques into a common library? If so, it would be worth arguing what the research contribution is.

The experiments consider 4 different neural networks from prior work that are evaluated on MNIST.
There are some experiments comparing different types of quantization, or different optimizers, but again it isn't very clear what to learn from this.
Given that the paper proposes a unified framework for various MPC techniques, I was expecting to see a much richer experimental section comparing different approaches. The paper claims that performing a wider range of comparisons might be very expensive. But I don't understand why these experiments have to be performed in "full" MPC. The impact of various MPC blocks on model accuracy could probably be easily assessed without performing an actual distributed training?

Overall, while the main experimental result in the paper (an MPC with >98% accuracy on MNIST) is quite neat, it appears that achieving this result mainly required the engineering effort of putting together the techniques from past works and actually running the experiments for long enough.
It is certainly cool to showcase this, but further details on the benefits of the proposed framework and additional experiments with it seem necessary before publishing this paper.

**Time Spent Reviewing:**

2

---

> ### Author Response · Authors · 2021-08-10
> **Response**
>
> We thank the reviewer for their feedback. We would like to point out that, despite a long list of papers conceptualizing full neural network training (including a variety of layers such as convolution), there has been no workable implementation other than the concurrent work of Tan et al. We strongly believe that the research doesn't stop at pointing out a possibility. Let us remark that secure computation is a different world, where even basic operations are tricky, and the main contribution of this paper is cross-discipline. It is not trivial, and it is not even done previously, to assemble the basic components into a basic deep learning algorithm (a convolutional neural network on MNIST) with reliable performance scores.

---

### Official Review · Reviewer_MTzN · 2021-07-19

**Rating:** 2
**Confidence:** 5

**Summary:**

The main contribution of this paper is an implementation of MNIST training using a three-party secure multi-party computation (MPC) protocol based on the MP-SPDZ library. The core claim is that full end-to-end training for 150 epochs can be completed in about 20 hours.

**Ethical Concerns:**

Not applicable.

**Limitations And Societal Impact:**

None of the limitations of the proposed implementation are discussed.

**Main Review:**

The authors themselves admit that all the secure computation building blocks described in Section 3 are already known individually. In fact, they also admit that all these building blocks are available in the high-level library that comes with the MP-SPDZ framework. Even on the machine learning side, the paper merely exploits existing network architectures reported in earlier papers without any modification or optimization. So the only contribution is to put the various pieces together to implement MNIST training and benchmarking the time taken with some previous publications. While this is indeed an useful effort that could be beneficial to the privacy-preserving machine learning community, the lack of any pretense of originality makes the paper unsuitable for publication. Unfortunately, the paper does not even attempt to analyze why the proposed implementation works faster, what are the bottlenecks, and what are the avenues for improvement.



**Time Spent Reviewing:**

2

---

> ### Author Response · Authors · 2021-08-10
> **Response**
>
> We thank the reviewer for their feedback. A few comments can be found below. We also address the comment regarding contribution in the response to reviewer Sgcg.
>
> Fast implementation: The used security model (three parties, one corrupt) is known to provide the most efficient security computation protocols. Furthermore, MP-SPDZ is a mature implementation whereas the comparable implementations have been created specifically for the related publications to the best of our knowledge.
>
> Bottlenecks: With MPC, both communication and local computation grow with the amount of secure computation. As our protocol does not rely on computation-heavy schemes such as homomorphic encryption, the main bottleneck is computation. Previous work has identified more complex mathematics such as exponentiations used for softmax as the bottleneck, but our work shows that following established machine learning algorithms relatively closely provides considerable benefits in terms of accuracy. For example, our work highlights that the extra cost for AMSgrad (roughly doubling) easily pays off in reaching certain accuracy levels far earlier.
>
> Avenues for improvement: The modularity of secure computation implies that any improvement in the building blocks results in an improved overall cost. However, the complexity of some building blocks such as our multiplication seems to match known lower bounds, which precludes further improvements. More promising is the extension to different security models such as a higher number of parties or corruption thresholds.

---

### Official Review · Reviewer_U3fp · 2021-07-21

**Rating:** 4
**Confidence:** 3

**Summary:**

The paper presents a secure multi-party computation (MPC) implementation of training several neural network architectures on MNIST. A 3-party semi-honest setup is considered with every individual party being honest-but-curious about the inputs of the other two parties. For a neural network with two convolution and two fully-connected layers, the paper reports the accuracy of 98.5% in 150 epochs of secure MPC training. The main contribution of the paper is to present an interesting combination of known MPC building blocks that are carefully selected from the literature.

**Limitations And Societal Impact:**

It will be helpful to discuss the limitations of the proposed MPC framework. Can it be scaled to larger number of parties and/or larger sized datasets? Which MPC components would be key bottlenecks to scalability in terms of computation or communication?

**Main Review:**

*Originality:* The MPC building blocks used in the paper are taken from the literature (and this is clearly stated by the authors). In other words, this work is an interesting combination of known MPC building blocks that are carefully selected from the literature. My main concern is that the comparison with the prior work is mentioned in a terse manner, and needs to be fleshed out.

1. As the authors mention, the main contribution of the paper is to put together secure computation building blocks in an efficient framework. Would it be possible to comment which building blocks are significantly different than the prior work ([Wagh et al. 2019, 2021], [Mohassel and Rindal 2018], [Tan et al. 2021])? Were some MPC components more critical than others in terms of accuracy/speed? Currently, it is not clear which components enable the authors to obtain superior results.

2. The authors mention that their implementation, which uses CPU, is 40% faster than [Tan et al. 2021], which uses GPU, speculating that communication costs could be the reason for this. Can the authors give more details on the communication cost comparison? The supplemental material has a table on communication per epoch, however, in some cases the proposed methods seem to use more communication than [Tan et al. 2021]. It will be helpful to include a detail comparison and a short discussion.

3. The authors mention that their implementation is built on MP-SPDZ [Keller 2020]. They also mention that “MP-SPDZ already featured capabilities to train dense neural networks…”. How does the proposed implementation differ? Which secure MPC building blocks are changed by the paper? It will be helpful to add more details.

*Quality:* The submission looks to be technically sound, but there are several avenues for improvement.

1. My main concern here is that the machine learning setup and the security model are not precisely defined. How is the dataset partitioned across parties? Is the trained model need to be kept secure as well (i.e., is the goal to compute secret shares of the trained model)?

2. Fig. 1 does not seem to convey sufficient technical details other than the high level idea. It is important to give more details.

3. The abstract claims the accuracy of 98.5% in 150 epochs for a specific neural network architecture. It will be helpful to include this case in Table 2 with more details to substantiate the claims.

*Clarity:* The paper can include more details on the building blocks and their implementation.

1. Instead of giving potential candidates (e.g., [Lu et al. 2020] is more direct than [Aly and Smart 2019] for inverse square root), it will be helpful to give more details/insights on the chosen component.

2. The current description of the security model is a bit confusing in my view. It is stated that “once party is expected to follow the protocol but might try extract information from their view”. Each of the three parties individually can do this, right? It will be helpful to add a formal description of the security requirements.

*Significance:* It is well-known that the problem of training neural networks using secure MPC is technically challenging. The paper seems to make progress on this problem. However, more details on the comparison with the prior work is necessary in order to understand the significance of the contributions.

**Time Spent Reviewing:**

6

---

> ### Author Response · Authors · 2021-08-10
> **Response**
>
> We thank the reviewer for their extensive feedback. Below we answer their questions, all of which we are happy to address in the paper.
>
> Originality:
> 1. The main difference to Wagh et al. is that they don't use softmax. As we note in the paper, they have not computed accuracy figures with their secure computation but a cleartext computation that uses a different activation function. Something similar applies to Mohassel and Rindal, who have used the ReLU-based activation function that was found to be inadequate by Keller and Sun (PPML'20). Tan et al. use a different approximation of exponentiation. Given that the other aspects are similar to ours, this might be the reason for their reduced accuracy. Regarding the differences in computation speed, we estimate that a considerable part can be attributed to the fact that MP-SPDZ is a more mature framework whereas comparable implementations have been created specifically for the related publications to the best of our knowledge.
> 2. We didn't mean to say that we improve on Tan et al. in terms of communication. Instead, we note that GPUs improve the cost of computation but not of communication, that is, data cannot be sent from a GPU to another host faster than from a CPU. In fact, there might even be a performance penalty because the communication infrastructure (encryption and operating system calls) usually runs on the CPU, not the GPU. MPC heavily relies on communication, which means that improvements in computation have a lesser impact on the wall clock time.
> 3. We have added the following components: backward propagation for 2D convolution and max-pooling, Dropout layer, Adam, and AMSgrad.
>
> Quality:
> 1. Our implementation is based on the black-box model of secure computation. This means that the parties input the data to the black box in exchange for a "handle" that is then used to execute the computation. In the context of federated learning, this means that the data can be partitioned in any way as long as there is a unique identifier for records because we don't run record linkage.
> 3. After fixing a bug post submission, we found that we can achieve 99% in five epochs and 99.2% in 25 epochs, which is within the range of the graphs.
>
> Clarity:
> 1. Our adaptation of Lu et al. is documented in detail. Can the reviewer give another example of a building block they would like more detail on? We found that the space limitation does not allow detailed descriptions of all algorithms.
> 2. The security is the usual threshold model with threshold one, which means that only one party is explicitly assumed to be corrupt. However, as the one party does not have to be fixed, it does imply that any single party can be corrupt.
>
> Limitations: Scaling to larger datasets implies a linear increase in computation cost as long as there is enough memory. Choosing another protocol does allow to increase the number of parties at a higher cost. A particular increase in cost would come from the probabilistic truncation, for which there is no known equivalent to the one used in our protocol.

---

### Official Review · Reviewer_VTuC · 2021-07-27

**Rating:** 6
**Confidence:** 4

**Summary:**

The authors train neural networks completely under an MPC setting and get accuracy results that are close to training in the clear. They utilize many techniques and tricks in a clever way, such as replicated secret sharing, domain conversion, quantization, oblivious selection and so on. They run experiments on MNIST and Fashion-MNIST datasets to find that training under an MPC setting slows down the training considerably by increasing the number of iterations required for convergence of the model with an accuracy of 98.5%.

**Limitations And Societal Impact:**

Limitations about high runtime are mentioned in the paper.

**Main Review:**

This paper presents an interesting combination of many well-known techniques and achieve secure neural network training with high accuracy. Authors compare their work with recent benchmarks and clearly explain how their work and results differ from theirs. The main limitation is that the protocol is implemented for three party computation and does not generalize to more parties. This might be a limitation as the main motivation for outsourced computations is to distribute the computation load. Moreover, no collusions between the parties are allowed. There is also the high runtime for high accuracy, but this is expected for an MPC-based setting. One related comment would be to have a dedicated section for the protocol and have a pseudocode of the algorithm in that section. The way the algorithm is presented is scattered and hard to follow.


**Time Spent Reviewing:**

6

---

> ### Author Response · Authors · 2021-08-10
> **Response**
>
> We thank the reviewer for their feedback. A few comments can be found below.
>
> - Generalization: The space limitation already allows only to give a high-level overview over a single protocol, which is why we chose to stick to one.
> - High runtime for high accuracy: After submission, we found a bug in our implementation that affected the accuracy (in the weight gradient computation of convolution back-propagation). We can now reach 99% in under an hour (5 epochs) and 99.2% in 3.5 hours (25 epochs).
> - Scattered presentation: Where are not sure what "the algorithm" refers to. Our presentation follows the modular nature of the implementation where higher-level building blocks are constructed from lower-level ones.

---

### Decision · Program_Chairs · 2021-09-27

**Decision:**

Reject

**Comment:**

This work demonstrates an implementation of MPC to train a convolutional neural network for MNIST that achieves 98.5 (now updated to 99.2) percent accuracy.

The reviewers were unconvinced that the contribution of the work is well-explained in the paper. The primary contribution is to bring together known techniques to demonstrate an efficient and high-accuracy implementation of neural network training with MPC. While I find this to be an impressive contribution, I would expect to see a more thorough empirical section comparing different approaches (potentially in simulation, as one of the reviewers said). In the current form, I do not find this paper to be a good fit for this conference.